



# Evaluating Weather and Chemical Transport Models at High Latitudes using MAGIC2021 Airborne Measurements

Félix Langot[1], Cyril Crevoisier[1], Thomas Lauvaux[2], Charbel Abdallah[2], Jérôme Pernin[1], Xin Lin[3], Marielle Saunois[3], Axel Guedj[1], Thomas Ponthieu[1], Anke Roiger[4], Klaus-Dirk Gottschaldt[4], and Alina Fiehn[4]

[1]Laboratoire de Météorologie Dynamique (LMD/IPSL), CNRS, Ecole Polytechnique, 91128 Palaiseau Cedex, France
[2]Groupe de Spectrométrie Moléculaire et Atmopshérique (GSMA), CNRS, Université de Reims-Champagne-Ardenne (URCA), 51100, Reims, France
[3]Laboratoire des Sciences du Climat et de l'Environnement (LSCE/IPSL), CEA/CNRS, L'Orme des Merisiers, Paris-Saclay, 91191 Gif-sur-Yvette Cedex, France
[4]Deutsches Zentrum für Luft- und Raumfahrt (DLR), Institut für Physik der Atmosphäre, Oberpfaffenhofen, Germany

**Correspondence:** Félix Langot (felix.langot@lmd.ipsl.fr)

**Abstract.** Methane ($CH_4$) fluxes emitted by wetlands at high latitudes remain one of the largest sources of uncertainties in global methane budgets. At these latitudes, flux estimation approaches, such as atmospheric inversions, are impacted by improper characterisation of atmospheric transport due to challenging meteorological conditions and a lack of measurements. Here, we assess the performances of ERA5 reanalysis, mesoscale simulations from WRF-Chem, and various atmospheric transport models from several global and regional inversion systems using meteorological and $CH_4$ in-situ measurements col-
lected during the MAGIC2021 campaign near Kiruna, Sweden. Over six measurements days in August 2021, ERA5 exhibited better agreement with observations than WRF-Chem thanks to data assimilation. Nevertheless, WRF-Chem demonstrated proficiency in simulating local atmospheric dynamics. Among global simulations of atmospheric concentrations of $CH_4$, inversion-optimised simulations of $CH_4$ concentrations yielded the best performances, particularly near the surface, with CAMS v21r1
marginally outperforming PYVAR-LMDz-SACS ensemble inversions. WRF-Chem regional simulations revealed performance disparities among $CH_4$ products, with positive biases in the boundary layer indicative of an overestimation of wetland emissions by selected wetland flux models. All transport models exhibited a vertically delayed gradient of $CH_4$ mixing ratios near the tropopause, resulting in a positive bias in the stratosphere. The high vertical resolution of CAMS *hlkx* facilitated a better representation of the vertical structure of $CH_4$ profiles in the stratosphere. Despite the limited spatiotemporal scope of MAGIC2021,
we were able to identify the best performing transport models and to evaluate fluxes from different biogeochemical model parametrisations using the MAGIC2021 high-resolution dataset.

## 1   Introduction

In recent years, the Earth's climate has been rapidly changing, with significant impacts on polar and sub-polar regions. In the Arctic, the rate of warming was thought to be around twice as fast as the global average until recently (AMAP, 2021;
Jansen et al., 2020; Walsh, 2014; Yu et al., 2021), but it is now estimated to be closer to 4 times faster (Rantanen et al.,



2022). The amount of greenhouse gas in the atmosphere and the meteorological conditions are essential components of the circumpolar climate system, where positive climate feedback loops are ubiquitous and disruptive (boreal fires (Zheng et al., 2023), permafrost, and wetland emissions(Zhang et al., 2023), albedo). However, scarcity of long-term direct observational data in the region has proven to be a challenge for studies aiming to constrain uncertainties and changes in the regional methane
cycle (Wittig et al., 2023). In order to understand these changes, climate models are therefore highly relied upon, and direct measurements must be employed to provide an assessment of their performance in modelling mixing ratios of greenhouse gases in the region.

In-situ data at high latitudes mainly come from several tall tower networks operated by Arctic countries, as depicted in Wittig et al. (2023). In Europe, data collection is coordinated by the Integrated Carbon Observation System (ICOS) network,
which comprises several towers stationed in Fennoscandia (few above the polar circle), that measure either in-situ atmospheric concentrations or methane fluxes through eddy covariance. Concentrations are however only measured close to the surface. They are mostly representative of local scales and lack vertical information. Measurements covering larger scales and higher atmospheric layers are crucial for accurately modelling the regional methane budget. Several projects have carried out field measurements of atmospheric methane at high latitudes recently, including campaigns from the NASA ABoVE initiative (Goetz
et al., 2011) or the NASA-ESA joint initiative Arctic Methane and Permafrost Challenge (AMPAC, Miller et al. (2021)). This latest project was notably involved in CoMet 2.0 Arctic (2022) set in Canada and Alaska, and MAGIC2021, set near Kiruna, Sweden (67 °N). The study presented here focuses on MAGIC2021, which spanned from 14 to 27 August 2021 and included measurements of atmospheric methane mixing ratios, combined with weather data sounding. The *Monitoring Atmospheric composition and Greenhouse gases through multi-Instruments Campaigns* (MAGIC) initiative launched by *Centre National de*
*la Recherche Scientifique* (CNRS) and *Centre Nationale des Études Spatiales* (CNES) aims at improving knowledge of $CO_2$ and $CH_4$ distribution and emissions in the Earth's atmosphere by organizing frequent measurement campaigns in different regions of interest. The first three campaigns, set in France from 2018 to 2020, served as a mean to calibrate and validate instruments and measurement techniques, whilst also validating current space missions. MAGIC2021 was therefore the first MAGIC campaign to focus on the study of $CH_4$ emissions at high latitudes, bringing together 70 participants from 17 teams
and 7 different countries. As field work is relatively recent, few results have been published yet and to our knowledge no study has tried to extensively assess atmospheric composition models using campaign data at high resolution in those regions.

Kiruna and its surrounding are characterised by wetland landscapes that include small ponds to large lakes as well as peatland and various inundated soils found in both boreal forest and tundra, as shown on Figure 1. These wetlands are known to be the main local source of methane though their emissions are generally poorly constrained (Saunois et al., 2020). Additionally,
some permafrost areas are also present at higher altitudes found in the Scandinavian mountains west of the city, though to a relatively small extent (Ahlenius, 2016). In lower parts of the atmosphere, model estimates of greenhouse gas mixing ratios can be strongly affected by these high uncertainties in emission processes, particularly for methane. Boundary layer mixing ratios are also strongly influenced by turbulent flow which is parametrised in global models and challenging to simulate accurately at finer scale (Schuh et al., 2019). This has a strong influence on atmospheric composition at all levels, as $CH_4$ released at
the surface is usually transported to deeper atmospheric levels via turbulent and/or convective fine scale processes. Above the





boundary layer, transport by geostrophic wind becomes the major driver for greenhouse gas concentrations. This means that atmospheric methane content is no longer strongly dependent on local emissions, but rather influenced by medium to long range transport. Stohl (2004) have shown that concentrations observed in Northern Europe can be traced back to emissions from North America or Siberia, provided meteorological conditions allowed for transport of surface emissions to the free

troposphere. At higher altitudes, the main driver of $CH_4$ mixing ratios becomes methane depletion by OH radicals. OH is found mainly at the top of the troposphere and at the bottom of the stratosphere, where other chemical species also react with $CH_4$, reducing drastically its presence above the tropopause. High-tropospheric and stratospheric $CH_4$ concentrations are therefore characterised by a strong vertical gradient. Tropopause height and troposphere/stratosphere exchanges become key influences on $CH_4$ mixing ratios there (Xiong et al., 2013), and are also challenging to model accurately (Mateus et al., 2022).

In this study, the accuracy and precision of several models in reproducing greenhouse gas mixing ratios and meteorological conditions observed at fine scale are assessed. Our study uses in-situ observations that employed research aircraft and weather balloons deployed around Kiruna in August 2021 (details in Section 2). We first start by assessing models regarding meteorological variables, with data from the European Centre for Medium-Range Weather Forecasts (ECMWF) fifth-generation reanalysis (ERA5) global product and regional WRF simulations. Then, we assess the atmospheric composition models ability

to reproduce observed $CH_4$ mixing ratios. Models assessed include the Copernicus Atmosphere Monitoring Service (CAMS) analysis *hlkx* and inversion-optimised flux product version 21r1, six PYVAR-LMDz-SACS ensemble inversions and WRF-Chem regional simulations. More detail about these models can be found in Section 2. Comparisons between model simulations and observational data provide insights into the strengths and limitations of these models in the Lapland region and highlight areas for improvement at several levels and scales.

## 75  2   Methods

### 2.1   Observational data

Both ground-based and airborne measurements were taken during MAGIC2021. This study focuses on airborne data taken by CNES weather balloons as well as SAFIRE ATR42 and DLR Cessna aeroplanes. These platforms had different payload configurations and measurement capabilities and thus provide complementary information about the distribution of gases in

the atmosphere. Whilst this study does not make use of the full set of MAGIC2021 measurements due to differences in ease and speed of data treatment specific to each instrument, it provides a solid example of such campaigns capability in terms of model validation.

### 2.1.1   Weather balloon observations

Two types of weather balloons from CNES were released during MAGIC2021: the Light Inflatable Balloon (BLD - *Ballon*

*Léger Dilatable*) and the Open Stratospheric Balloon (BSO - *Ballon Stratosphérique Ouvert*). Balloon types differ in their usage, BLD are single-use, their membrane bursting after the ascent phase. They typically reach altitudes up to 30km. BSO



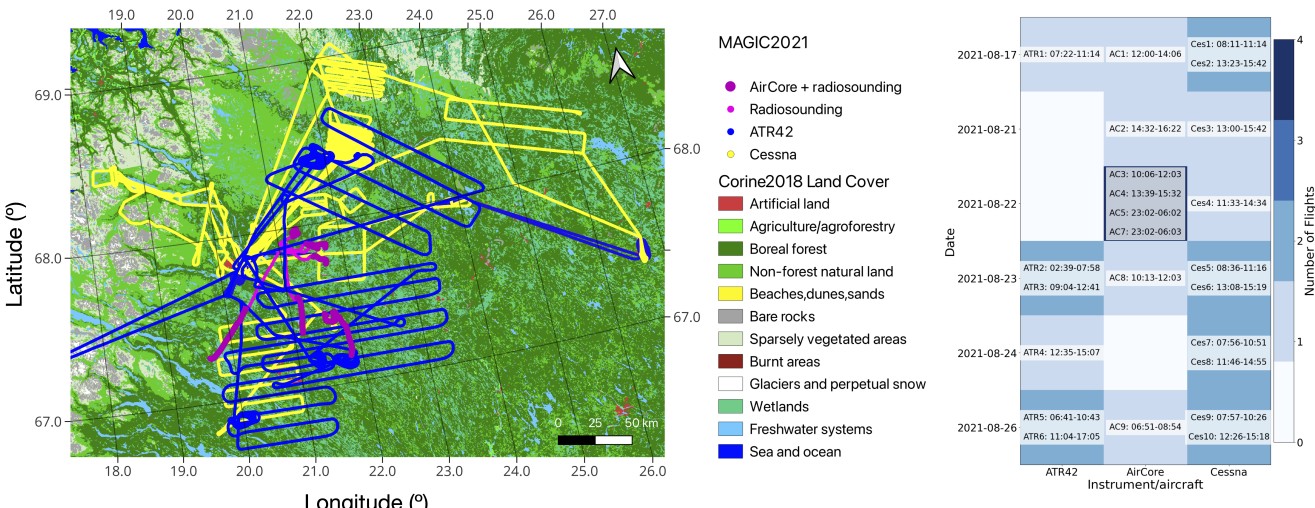

**Figure 1.** Location (left) and date and time (right) of MAGIC2021 measurements used in this study. The left panel does not show the last ATR42 trajectory of measurements as it left the region of Kiruna to fly over oil platforms off the Norwegian coast.

are reusable and can reach altitudes above 30km. Weather balloons carried two main instruments whose data were used in the study: the AirCore atmospheric sampler, and meteomodem M20 radiosondes. The M20 instrument is an ultra-lightweight (36 grams) radiosonde used to gather meteorological data such as temperature, humidity, pressure, as well as zonal (U) and meridional (V) wind components. More details about the instrument can be found at Meteomodem (2020). Measurements with the M20 were made during both ascending and descending phases of balloon flights.

AirCore is an atmospheric sampler (Tans, 2009; Karion et al., 2010; Membrive et al., 2017) which allows sampling atmospheric composition on a large range of altitudes (∼0 - 30km) using the atmospheric pressure gradient. Both light AirCore and high resolution AirCore were deployed. After sampling, AirCores were retrieved and analysed on the ground, using cavity ring-down spectroscopy (CRDS). More specifically, analysis of sampled air was performed by two models of spectrometers. The G2401 and G5310 instruments from Picarro© (Picarro, 2008). The G2401 model measured $CH_4$, $CO_2$, and $H2O$, whilst the G5310 measured CO. Time and trajectories of measurements for the AirCore instrument are shown in Figure 1

### 2.1.2 Aircraft observations

Two research aircraft flew between the surface and approximately 8 km, carrying instruments that gathered atmospheric composition and weather data. In this study we used data from two aircraft: the SAFIRE ATR 42-320 (CNES, CNRS, Météo France), abbreviated as ATR42, and the DLR Cessna C-208B Grand Caravan (DLR), abbreviated as Cessna. The position, velocity, and altitude of the ATR42 aircraft were recorded by both an iXBlue™ inertial reference/navigation system called SAFIRE AIRINS and a NovAtel™ Global Positioning System (GPS). This GPS system consists of L1/L2 GPS-Antennae (5x) and a OEM3 receiver. Water vapour and relative humidity were measured using a non dew/frost point hygrometer called



SAFIRE relative humidity sensor, made by Michell Instruments™. Airspeed, incidence angle and turbulence were measured by a Rosemount & Sextant™ incident flow vector probe called SAFIRE five hole radome. This instrument allows to measure U and V wind components. Finally, the Rosemount™ in-situ temperature sensor called SAFIRE Rosemount PT102E2AL, measures the temperature at the aircraft's location. Also on board the ATR42 were two Picarro™ models previously mentioned that were used for in-situ atmospheric composition analysis, as well as several other instruments distributed on the aircraft that gathered meteorological data.

The Cessna aircraft was equipped with a system called blackMAMBA (Measurement Acquisition of Meteorological Basics) that delivered track (i.e. position and time) data, together with aircraft status and meteorological parameters. Some of the meteorological sensors were installed in the MetPod, a container with a nose boom, mounted under the left wing. This allows atmospheric parameters to be measured with less distortion than if they were measured from the fuselage. The temperature, pressure, humidity sensors and the calibration of the wind measurement system are described in detail by (Mallaun et al., 2015). The aircraft also carried two in-situ trace gas instruments. Here we use only the data from a Picarro G1301m, which measured $CH_4$, $CO_2$, and $H_2O$ mixing ratios. More details about gas measurements can be found in Fiehn et al. (2020).

Observations used in this study therefore contain 8 separate weather balloon samplings, 6 ATR42 and 10 Cessna flights for the atmospheric composition. For meteorological data, only 6 of the 8 weather balloons were used due to radiosondes malfunctioning during two of the flights, but meteorological data was measured during both ascent and descent flight phases which allowed for a significant

## 2.2 Atmospheric modelling systems

This section describes model data that was compared to MAGIC2021 observations. The first two sections describe global models whilst the third focuses on the regional modelling system based on WRF-Chem that was especially set up for MAGIC2021.

### 2.2.1 Global meteorological reanalysis

Global meteorological fields used in this study came from the European Centre for Medium-Range Weather Forecasts (ECMWF) fifth-generation reanalysis product (ERA5, Hersbach et al. (2020); C3S (2018)), that provides meteorological data on a global scale from 1950 to present. In our study, we assessed ERA5 reanalysis wind, temperature, and humidity. The higher density of vertical levels in ERA5 from the mid-troposphere down to ground level allows for accurate comparison with the flights from MAGIC2021. Our analysis was carried out using ERA5 at time resolution of 1 hour, spatial resolution of 0.25°and 137 vertical levels. Horizontal ERA5 wind was given in terms of zonal (U) and meridional (V) components of the wind vector. Both observations and model data were converted to wind speed $W_{spd}$ and direction $W_{dir}$ for comparison when needed using: $W_{spd} = \sqrt{U^2 + V^2}$ ; $W_{dir} = \tan^{-1}(-U, -V) \cdot \frac{180}{\pi}$ (Tetzner et. al 2019). To compare humidity from observations, that measured relative humidity (RH), to ERA5 humidity, given as specific humidity $q$, ERA5 data was converted to RH using RH $= \frac{e}{e_s}$ where $e$ is the partial pressure of water vapour in air (pressure exerted by water molecules) and $e_s$ is the saturation vapour pressure, or the maximum vapour pressure that can occur at a given temperature before condensation occurs.



### 2.2.2 Global CH$_4$ assimilation systems

The Copernicus Atmosphere Monitoring Service (CAMS) is a service provided by ECMWF. Its atmospheric composition product combines satellite data and ground-based measurements in a 4D-Var assimilation system to provide comprehensive

information on key atmospheric parameters such as mixing ratios of greenhouse gases in 4 dimensions (Peuch et al., 2022). Two CAMS products are used in this study. The first is the CAMS *hlkx* analysis (Agustí-Panareda et al., 2023) which is based on ECMWF Integrated Forecast System for Composition (C-IFS, Verma et al. (2017)), with a vertical resolution of 137 vertical levels, a horizontal resolution of 0.25°and 6 hours of temporal resolution. Methane loss to OH in the upper troposphere and stratosphere is provided by Bergamaschi et al. (2009) where CH$_4$ destruction was simulated using OH fields based on methyl

chloroform optimised Carbon Bond Mechanism 4 (CBM-4) chemistry (Bergamaschi et al., 2005; Houweling et al., 1998). Non-OH stratospheric loss is based on the 2-D photochemical MaxPlanck-Institute (MPI) model (Brühl and Crutzen, 1993).

The second CAMS product compared to MAGIC2021 is the global inversion-optimised greenhouse gas concentrations product for CH$_4$ version 21r1 (Segers, 2023). This product makes use of methane concentration measurements from the NOAA ground observations network to optimise a priori fluxes of CH$_4$ and produce 3D concentrations and correspond better to

ground observations. Simulations are run using the chemistry transport model TM5-MP (Williams et al., 2017) that includes upper tropospheric and stratospheric computation of CH$_4$ loss using monthly concentrations of sink tracers, built-in reaction rates and monthly temperature estimates. Tropospheric or stratospheric reaction rates are attributed using a latitude dependent tropopause parametrisation from Lawrence et al. (2001). The spatial resolution is of 3°×2°×34 levels and a temporal resolution of 6 hours. To distinguish between these two products from CAMS, the analysis product will be referred to as CAMS *hlkx* and

the inversion-optimised product as CAMS v21r1.

Campaign data was also compared to concentrations from six PYVAR-LMDz-SACS (abbreviated PLS) ensemble inversions that optimised weekly methane surface fluxes from 2019 to 2021 at a spatial resolution of 1.9°× 3.75°(latitude× longitude) on 39 vertical levels and 3-hourly time resolution. Inversions employed three different atmospheric observation datasets for flux constraints and two physical parametrisations. Two inversions used GOSAT column estimates to constrain fluxes, either

from the National Institute for Environmental Studies (NIES) or University of Leicester (UoL) and the others used surface observations from both the ICOS and NOAA tower networks. The two physical parametrisation are known as the classic and standard versions of the atmospheric transport model LMDz (noted a and b respectively). The classic version uses the vertical diffusion scheme of Louis (1979) and the scheme of Tiedtke (1989) to parametrise deep convection, whilst the standard version combines the vertical diffusion scheme of Mellor and Yamada (1974) and thermal plume modelling by Rio and Hourdin (2008)

to simulate the atmospheric mixing in the boundary layer. Deep convection is represented using Emanuel (1991) scheme coupled with the parametrisation of cold pools developed by Grandpeix et al. (2010). Bottom-up inventories or process-based land surface models were used to build prior CH$_4$ fluxes for different categories, and the OH and O($^1$D) fields were prescribed from the simulation of a chemistry-climate model LMDz-INCA with a full tropospheric photochemistry scheme. Inclusion of observations and definition of observation errors to constrain fluxes followed the method outlined in (Peng et al., 2022; Lin

et al., 2023).





### 2.2.3 Regional atmospheric model (WRF-Chem)

*WRF-Chem configuration*

In addition to global model outputs, the Weather Research and Forecasting coupled with Chemistry (WRF-Chem) model was used to simulate the meteorological conditions and greenhouse gas concentrations during the MAGIC2021 campaign on a
regional scale. WRF is a widely used mesoscale numerical weather prediction system in both research purposes and operational forecasting. It uses fully compressible and non-hydrostatic Eulerian equations on an Arakawa C-staggered grid to ensure the preservation of mass, momentum, entropy, and scalars (Skamarock et al., 2008). The set-up for this study included two domains, one parent and one nested. The parent domain (d01) encompassed the whole of Fennoscandia as well as Denmark, the westernmost part of Russia and most of the area covered by Baltic countries, at a resolution of 9×9 km. The nested
domain (d02) had a higher resolution of 3×3 km and spanned most of the northern part of Finland, Sweden and Norway where MAGIC2021 measurements were taken. Domain boundaries were chosen such as to avoid strong emissions and high topography close to a boundary, which are known to cause transport problems (NCAR, 2024). WRF-Chem generated output fields including meteorological variables and concentrations every 20 minutes.

Our physical parametrisation included the WSM5 scheme for microphysics (Hong et al., 2004) as well as the RRTMG
longwave and shortwave schemes (Iacono et al., 2008) for radiation. The planetary boundary layer was represented using the MYNN Level 2.5 scheme (Nakanishi and Niino, 2009), whilst the revised MM5 surface layer scheme (Jiménez et al., 2012) was used, with the thermal roughness length dependent on vegetation. No urban model was activated. For the land surface, the Noah model was used, with 4 soil layers (Tewari, 2004). Regarding convection, the Kain-Fritsch scheme was used for the parent domain (Kain, 2004), whilst convection was resolved explicitly in the nested domain. Additional convection-related options
were activated, including radiation feedback on convection, convection diagnostics, and Grell-Devenyi scheme parameters (Grell and Dévényi, 2002). Vertically, the simulations had 50 levels from ∼140m to ∼20km with about half of all levels below 2km. The model configuration was evaluated in previous studies to produce minimum transport errors at both continental (Feng et al., 2019) and regional (Díaz-Isaac et al., 2018) scales.

Methane concentrations were modelled as passive tracers, which were transported online at each time step concurrently with
meteorological variables. Emissions are injected from the surface into the first atmospheric layer to generate the concentration fields of tracers. These tracers undertook a series of transport processes, including advection, diffusion, turbulence, and convective mixing, to simulate the motion of molecules in the atmosphere. Initial conditions were set by ERA5 reanalysis meteorology at 0.5°×0.5°×137 levels resolution and boundary meteorological conditions were updated every 3 hours using the same product. Data within WRF-Chem domain was then produced by WRF physics and dynamics. Methane boundary condi-
tions were produced by the inversion optimised CAMS concentrations product version 21r1 described earlier, at a resolution of 3°×2°×34 levels every 6 hours. Emissions within simulation domains were divided into multiple tracers depending on source types. These tracers are described in Table 1. Concentrations within our simulation domain were initially set to a constant value. A period of 15 days was shown to be sufficient for boundary conditions and local emissions to propagate through our





domains and reach steady-state. The simulations were thus run from 01/08/2021 to 31/08/2021, to account for spin-up time
and the MAGIC2021 campaign period.

| Source | Model | Spatial resolution | Time resolution |
|---|---|---|---|
| Anthropogenic | CAMS | 0.1°×0.1° | monthly clim. (2016-18) |
| Fire | CAMS | 0.1°×0.1° | daily aug. 2021 |
| Oceanic | Weber et al. (2019) | 0.25°×0.25° | monthly clim. (1980-2016) |
| Wetland | WetCHARTs | 0.5°×0.5° | monthly clim. (2016-18) |
| Wetland | JSBACH-HIMMELI | 0.1°×0.1° | daily aug. 2021 |
| Lakes | Johnson et al. (2022) | 0.25°×0.25° | daily clim. 2003-2015 |

**Table 1.** Emission sources used in the WRF-Chem simulations, given with spatial and temporal resolutions.

*Emission tracers*

Input emissions were chosen according to data availability for August 2021, then prioritising higher spatial resolution in
order to reduce regridding issues. If no product were found for that time period, the highest time resolution product was chosen
and climatological averages were used.

Oceanic methane emissions were taken from Weber et al. (2019), a monthly climatology with a spatial resolution of
0.25°×0.25°. Methane lake emissions from Johnson et al. (2022) were also used. The dataset includes corrections for daily
and seasonal observational bias, observed ice-free/emission seasonality, and realistic lake area and distribution. Anthropogenic
and fire emissions of methane were provided by CAMS, which publishes emissions driving their global atmospheric green-
house gas concentrations products (Agustí-Panareda et al., 2023). They are respectively from EDGARv4.2FT2010 (Olivier
and Janssens-Maenhout, 2012) and GFAS Version 1.2 (Kaiser et al., 2012). Anthropogenic and fire emissions both share the
same 0.1°×0.1°spatial resolution but anthropogenic emissions were monthly averaged emissions over 2016-2017-2018 (latest
years available) whereas fire emissions were daily emissions from August 2021. Wetland emissions came from two sources:
the latest product from the WetCHARTs model (Bloom et al., 2017), with simulations up to 2019, and several versions of
JSBACH-HIMMELI (JSB-HIM) simulations originally designed for the European project VERIFY, described in Aalto (2019),
that were recently extended to later years. WetCHARTs has a spatial resolution of 0.5°×0.5°and a monthly time resolution,
spanning until 2019. A monthly climatological average was therefore used, taking the same years as for CAMS anthropogenic
emissions. 18 different flux versions are publicly available, depending on physical parameters detailed in the documentation
(Bloom et al., 2017). A subset of 8 WetCHARTs versions were selected, to maximise representativeness of the dataset whilst
staying cost-effective in our computations. JSB-HIM emissions were provided by the Finnish Meteorological Institute (FMI)
at daily resolution for August 2021 and a spatial resolution of 0.1°×0.1°. 3 versions of total wetland flux differing in their
driving meteorology were included in this study.





Inventory emissions all have different spatial resolution, so they have to be regridded to our WRF-Chem domains resolution. This was done by interpolating emissions from our data products to the WRF-Chem grid (Virtanen, 2010). 11 emission tracers were dedicated to regional CH₄ emissions An additional tracer was dedicated to boundary conditions. These were provided

by the inversion-optimised CAMS v21r1 product described in Section 2.2.2 and interpolated onto WRF-Chem vertical levels using Lauvaux (2022). Additionally, artificial boundary conditions were also implemented for other tracers in order to prevent near-zero computation error propagation throughout the whole simulation. This was done by hourly adding a constant offset of 300ppb through the emission tracers domain boundaries. WRF-Chem supports several independent passive tracers. This allows us to construct different versions of atmospheric methane concentrations from a single simulation. A common core of

methane concentrations was built using the boundary condition tracer added to the sum of anthropogenic, fire, oceanic and lake emissions tracers. To this common core, wetland contributions can be separately added to obtain different atmospheric methane concentrations. These wetland emissions include 8 separate products from the WetCHARTs inventory, and 3 products from JSB-HIM simulations as described above (Bloom et al., 2017; Aalto, 2019). Simulations were run in both d01 and d02 domains, resulting in a total of 22 atmospheric CH₄ concentrations product.

## 2.3 Comparison method

### 2.3.1 4 dimensional barycentric interpolation using Delaunay triangulation

In our comparisons, modelled data were interpolated on measurement locations using the python function `scipy.interpol-ate.griddata` from the scientific python library `scipy`. The function `griddata` uses `scipy.interpolate.Linear-NDInterpolator` when performing linear interpolation in multiple dimensions as in our case, a function that was written in

cython by Virtanen (2010). Interpolation is necessary because gridded modelled data do not have the same temporal or spatial resolution as measurements taken by balloons or aircraft. Additionally, using Delaunay triangulation as in `griddata` allows interpolation from an irregular grid such as the pressure grid used in studied models. The interpolation was performed in 4 dimensions (time + 3 space dimensions). `griddata` first computes a Delaunay triangulation around the measurement coordinates to pick out interpolating points from the model grid. In 4 dimensions, each simplex around an observation point contains

5 vertices corresponding to 5 model coordinates in 4D. Barycentric linear interpolation is then performed using each simplex's 5 vertices to compute a model value at a particular measurement location. This method enables a fast, easy to implement and accurate comparison between modelled and measured data, by allowing comparison along each instrument's individual trajectory.

### 2.3.2 Statistics

Our analysis systematically divided comparisons in 3 layers: surface (P>800 hPa = BL), mid-tropospheric (300<P<800 hPa = FT) and top of troposphere/bottom of stratosphere (P<300 hPa = UTLS). BL was chosen as such to incorporate the boundary layer for all the field measurements period. P<300 hPa was chosen as it corresponds to the height at which chemical reactions





and exchange processes between stratosphere and troposphere start to strongly affect methane concentrations. These values were picked as constants to ease our calculations. The contribution of each instrument to these layers is shown in Table 2.

| Aircraft | Share of all sample (%) | | Share of BL sample (%) | | Share of FT sample (%) | | Share of UTLS sample (%) | |
|---|---|---|---|---|---|---|---|---|
| | CH$_4$ | Meteo | CH$_4$ | Meteo | CH$_4$ | Meteo | CH$_4$ | Meteo |
| AirCore | 3.94 | 16.98 | 0.83 | 2.36 | 6.16 | 14.74 | 100.0 | 100.0 |
| ATR42 | 46.22 | 39.94 | 21.16 | 20.83 | 93.56 | 85.0 | 0.0 | 0.0 |
| Cessna | 49.84 | 43.08 | 78.01 | 76.81 | 0.28 | 0.26 | 0.0 | 0.0 |

**Table 2.** Contribution of MAGIC2021 aircraft and balloons to sample data. CH$_4$ measurements were only made on the descending phase of the AirCore flights, and 2 of the 8 balloons flights were not used because of an instrument failure as mentionned in Section 2.

Four statistics were computed to assess model performance against observations in each of the three previously defined layers and to compare the performance of models. These were namely the mean difference (model - observation) between measured physical quantities and interpolated model quantities over a given sample $\overline{\Delta}$, standard deviation $\sigma$, Pearson correlation $\rho$, and root-mean-square error RMSE. Circular statistics from Mardia (1972); Jammalamadaka and Sengupta (2001) were applied to compare wind directions by computing circular $\overline{\Delta}$, $\sigma$, $\rho$ and RMSE associated with model and observed directions.

These statistics were used to draw Taylor diagrams (Taylor, 2001) which allow to assess a set of models against observations. These diagrams cleverly combine $\rho$, $\sigma$ and centred RMSE (CRMSE) in a polar coordinate plot using the law of cosines. The radial coordinate of a data point usually represents the standard deviation ($r = \sigma$) whilst angular position gives its correlation with observations ($\theta = \arccos(\rho)$). A reference point is set at ($\sigma_{\mathrm{obs}}$, $\rho_{\mathrm{obs}}$) where $\sigma_{\mathrm{obs}}$ is the standard deviation of the observations and $\rho_{\mathrm{obs}} = 1$. Here we normalise $\sigma$ to be able to compare quantities from different layers of the atmosphere onto the same plot:

$\sigma_{\mathrm{N}} = \sigma/\sigma_{\mathrm{obs}}$. The coordinates of the reference point become (1,1). The better the model, the closer to this reference point it will be. CRMSE can also be represented on the diagram, as the radial distance from the reference point. Taylor (2001) shows:

$$\mathrm{CRMSE} = \sqrt{\frac{1}{N}\sum_i^N \left[ \left(x_i^{\mathrm{obs}} - \overline{x^{\mathrm{obs}}}\right) - \left(x_i^{\mathrm{mod}} - \overline{x^{\mathrm{mod}}}\right)\right]^2} = \sqrt{\sigma_{\mathrm{obs}}^2 + \sigma_{\mathrm{mod}}^2 - \rho\sigma_{\mathrm{obs}}\sigma_{\mathrm{mod}}}$$

This statistic is a measure of model spread around observational values after removing any bias. It is therefore useful to quantify model noise but it lacks an assessment of distance between model estimates and observations. To remedy this, we chose to pair each Taylor diagram with a plot of RMSE against $\overline{\Delta}$ as in Kärnä and Baptista (2016).

**3   Weather data comparison results & Discussion**

**3.1   Winds**

Figure 2 considers both wind speed and directions as observed during MAGIC2021 and modelled by ERA5 and WRF. In the lowest analysis level (BL), MAGIC2021 winds showed 3 main directions of origin: SSW (18%), N (17%) and S (15%) which







**Figure 2.** Wind rose plots for MAGIC2021 observed and ERA5 modelled winds interpolated on the flight tracks of the measurement platforms. Observations include ATR42, AirCore and Cessna measurements. The radial axis gives the proportion (in %) of winds coming from a given direction given by the angular axis. Coloured bins represent speeds associated with each direction.



accounted for half the of measured winds. Their speeds ranged from 2 to 12 m/s. Both ERA5 and WRF also showed a large
(20%) contribution of SSW wind and distribution of wind speeds similar to observations. WRF showed a stronger contribution
of N winds compared to observed winds in both domains (30% vs 16%), whilst ERA5 seems too overestimate the NNE wind
contribution (16% vs 10%).

In our mid-tropospheric layer (300<P<800 hPa), measured winds showed two main directions of origin: N (27%) and NNE
(26%). A large share of these winds had speeds higher than 16 m/s (60km/h). Wind directions and speed distribution of both
models agreed very well with observations in this layer both in terms of direction and speed, with more than 20% of N and
NNE winds for all simulations. ERA5 had NNE contributions more important than N wind, contrary to observations and WRF.
However it showed a distribution of wind speeds closer to observations than WRF, which had a more important share of low
speed winds than observations. WRF winds were again very similar between the two domains. They were overestimating the
contribution from NNW, especially of low speed winds.

Measured winds at P<300 hPa showed 5 main directions of origin: 4 westerly (W, WSW, SW and WNW) and 1 from NNE.
The former showed a speed distribution characterised by low values ranging from 2 to 10 m/s, whilst the latter had a majority
of its speeds >16 m/s. Both models also simulated an important contribution from westerly winds at low speeds (>50% for
observations and all models). WRF showed a tendency to overestimate contributions from main components and underestimate
those from secondary origins in both domains, giving for example a higher contribution from NNE winds than in ERA5 and
observed winds, but no contribution to S or ESE directions. ERA5 showed an opposite behaviour, as all contributions were
more evenly distributed than in observations.

We now look at the statistical performance of these models in terms of wind speed and direction separately.

Figure 3 shows ERA5 correlated better than WRF with MAGIC2021 wind speeds in the bottom two layers, and had a $\sigma_N$
value closer to 1 than WRF in those layers as well. This latter statement was also true in the upper atmospheric layer but only
marginally so. Both WRF-Chem domains showed marginally better correlation there, with d02 being slightly better than d01.
In terms of RMSE and $\overline{\Delta}$, ERA5 also performed better than WRF in two of the 3 studied layers, but was outperformed by
WRF in terms of $\overline{\Delta}$ in the lowest (P>800 hPa) layer whilst performing similarly in terms of RMSE. $\rho$ & $\sigma$ performance was
better with increasing altitude for all model. In terms of RMSE and $\overline{\Delta}$, this was however not the case. WRF performed best in
the lowest layer, then in the highest layer and had its worst performance in the free troposphere. ERA5 had its lowest $\overline{\Delta}$ and
its highest RMSE in the middle layer. Going from FT to UTLS, ERA5 $\overline{\Delta}$ changed sign and increased in magnitude whilst the
RMSE was slightly reduced. This is also the case going from UTLS to BL, with ERA5 $\overline{\Delta}$ being more negative and RMSE
being reduced again. Notably, WRF coarser domain results were closer to observations in 3 of the 4 statistics studied here ($\sigma_N$,
$\overline{\Delta}$ & RMSE) than the finer domain, whilst performing similarly in terms of $\rho$.

ERA5 and WRF also showed good agreement with MAGIC2021 wind directions both in terms of $\rho$ (0.7<$\rho_{mod}$<0.96) and
$\sigma_N$ (0.6<$\sigma_{Nmod}$<1.1), as shown in Figure 3. UTLS was the only layer with a significant gap in performance between WRF and
ERA5, where WRF had $\sigma_N$ similar to ERA5 but a higher correlation ($\rho_{ERA5} \approx 0.7$ vs $\rho_{WRF} \approx 0.95$). Again both WRF-Chem
domains had close performance, mostly in UTLS. In BL, the closest model to observations in terms of $\sigma_N$ was WRF-Chem
d01, followed by WRF-Chem d02 and ERA5. ERA5 had the best correlation with observations in BL, followed by WRF-

**Figure 3.** Taylor and RMSE/$\overline{\Delta}$ diagrams for wind speed (top) and direction (bottom). Taylor diagrams (left) radial axis represents the normalised standard deviation of modelled wind speeds/directions. The angular axis represents correlation between modelled and observed wind speeds/directions. Centred RMSE is represented by the radial distance from the reference point. RMSE versus $\overline{\Delta}$ (right) for wind speed/direction comparisons between MAGIC2021 observations and modelled winds (in m/s and ° respectively).





Chem d01 and WRF-Chem d02. WRF-Chem d02 and ERA5 had best performance in terms of $\sigma_N$ in FT, with ERA5 data

being better correlated with observations. Looking at $\overline{\Delta}$ and RMSE, all models performed well with $\overline{\Delta}$ being all positive and

$\lesssim 6°$throughout all 3 layers. Performance of WRF-Chem d01 was noticeably homogeneous across the 3 layers, having a RMSE

just under 2°and $3 < \overline{\Delta} < 4°$. Best model performance was by WRF-Chem d02 and ERA5 in FT and UTLS respectively, with

both RMSE and $\overline{\Delta}$ below 1.

### 3.2 Temperature

Figure 4 shows $\overline{\Delta}$ profiles of temperature for ERA5, WRF-Chem d01 and WRF-Chem d02 computed against the three

MAGIC2021 datasets: Cessna, ATR42 and AirCore. In BL, temperature $\overline{\Delta}$ variations were similar for all models. Compar-

ison with ATR42 and Cessna data showed a $\overline{\Delta} \approx 0$ whilst AirCore comparisons showed a clear negative $\overline{\Delta}$ ranging from 0 to

5K throught this layer. In FT, profile data showed $T_{ERA5} < T_{WRF}$ with ERA5 being negatively biased when compared to weather

balloon data and WRF being positively biased when compared to ATR42 data. Finally, weather balloon data for UTLS showed

a good agreement between both ERA5, WRF and measured T, but with more variation around $\overline{\Delta}T = 0$ for WRF. WRF values

cannot be compared to weather balloon data in UTLS above P≈50 hPa as this was set as the upper limit of the model domain.

Looking at Figure 6, it can be seen that all models performed very well in every layer, being all close to $\sigma_N = 1$ and

correlating very well with observations ($\rho > 0.9$). In terms of RMSE and $\overline{\Delta}$, there was a clear gap in performance for all

three models when looking at BL versus FT & UTLS, as a clear negative temperature $\overline{\Delta}$ was observed in BL, showing also

a relatively high RMSE when compared to FT & UTLS. Although the gap was small, it is worth noting again that ERA5

performed better than WRF in terms of correlation, $\overline{\Delta}$ and RMSE in all layers, and it was only in BL that ERA5 showed a $\sigma_N$

value slightly further from 1 than WRF, as performance was otherwise close regarding that metric. Overall, WRF-Chem d01

and d02 showed closely similar performance in all layers.

### 3.3 Humidity

$\overline{\Delta}$ profiles of relative humidity for ERA5, WRF-Chem d01 and WRF-Chem d02 were computed, as shown in Figure 5. BL pro-

files suggest $RH_{ERA5} > RH_{WRFd01} > RH_{WRFd02}$ with WRF d02 being closer to $\overline{\Delta} = 0$ throughout the layer. Below the tropopause

(~13 km), $\overline{\Delta}$ profiles were relatively noisy, with $\overline{\Delta}$ ranging from about -20 to 20% in BL and UTLS, and from about -30 to

25% in FT. This was not surprising as RH was noisy in observational data. Appart from this, all models performed well in

terms of $\overline{\Delta}$ throughout BL, FT and UTLS, with $\overline{\Delta}$ being close to 0 and each model showing similar $\overline{\Delta}$ variations along their

profiles. ERA5 showed a slightly more positive $\overline{\Delta}$ at the bottom of FT than both WRF simulations. In UTLS, the profile was

cut at P = 100 hPa as RH drops rapidly to values ~0 in the stratosphere where $\overline{\Delta}RH$ should be studied on a different scale, and

it was deemed to be beyond the scope of this study.

Figure 6 shows that models correlated best with MAGIC2021 measurements in UTLS, followed by FT and BL. Variability

was however better represented in BL, with $\sigma_N$ being closer to 1 than in FT and UTLS for all 3 models. Model performance

was good overall, but showing worst numbers than for temperature. $\sigma_N$ values ranged from 0.65 to 1.15 and $\rho$ went from just

under 0.6 in BL to ~0.95 in UTLS. ERA5 performed once again better both in terms of correlation and $\sigma_N$ than WRF in all





**Figure 4.** Temperature intercomparison between MAGIC2021 data and weather models. Profiles are computed using full weather balloon dataset and profile sections of ATR42 and Cessna flights. ERA5 related profiles are shown in green, WRF d01 in blue and WRF d02 in red. **Top:** Mean temperature profiles accross all flights from each aircraft (black) plotted with modelled temperatures interpolated on aircraft trajectories: Cessna (left), ATR42 (centre) and weather balloon (right). **Middle:** Vertical profile of mean temperature difference between measurements and models for each platform: Cessna (left), ATR42 (centre) and weather balloon (right). $\Delta\mathrm{T} = \overline{\mathrm{T_{mod}} - \mathrm{T_{obs}}}|_{\mathrm{L}}$ for atmospheric level L. **Bottom:** Sections of mean temperature difference vertical profiles correponding to the 3 analysis levels: BL (left), FT (centre) and UTLS (right), where Cessna data is shown in dashed lines, ATR42 data in dotted lines and AirCore data in solid lines. Shaded areas represent the $1\sigma$ deviation from the mean temperature or temperature difference profiles.





**Figure 5.** RH intercomparison between MAGIC2021 data and weather models. Profiles are computed using full weather balloon dataset and profile sections of ATR42 and Cessna flights. ERA5 related profiles are shown in green, WRF d01 in blue and WRF d02 in red. **Top:** Mean RH profiles accross all flights from each aircraft (black) plotted with modelled RH interpolated on aircraft trajectories: Cessna (left), ATR42 (centre) and weather balloon (right). **Middle:** Vertical profile of mean RH difference between measurements and models for each aircraft: Cessna (left), ATR42 (centre) and weather balloon (right). $\Delta T = \overline{RH_{mod} - RH_{obs}}|_L$ for atmospheric level L. **Bottom:** Sections of mean RH difference vertical profiles corrreponding to the 3 analysis levels: BL (left), FT (centre) and UTLS (right), where Cessna data is shown in dashed lines, ATR42 data in dotted lines and AirCore data in solid lines. Shaded areas represent the $1\sigma$ deviation from the mean RH or RH difference profiles.







**Figure 6.** Taylor and RMSE/$\overline{\Delta}$ diagrams for temperature (top) and relative humidity (bottom). Taylor diagrams (left) radial axis represents the normalised standard deviation of modelled T/RH. The angular axis represents correlation between modelled and observed T/RH. Centred RMSE is represented by the radial distance from the reference point. RMSE versus $\overline{\Delta}$ (right) for T/RH comparisons between MAGIC2021 observations and modelled winds (in °C and % respectively).



layers. Looking at $\sigma_N$ changes along the vertical, models show a that variability increases with pressure than the one observed, with UTLS $\sigma_N$ < FT $\sigma_N$ < BL $\sigma_N$ for all models. In terms of RMSE and $\overline{\Delta}$, RH was least well represented in FT, where a negative $\overline{\Delta}$ of ~3 % was observed for all models. This is also where RMSE was highest, ranging from 13 to just below 17 %.

An opposite $\overline{\Delta}$ was observed in BL, of a similar magnitude for ERA5 and WRF-Chem d01, but lower for WRF-Chem d02, where $\overline{\Delta}$<1%. In UTLS, all models showed a small negative $\overline{\Delta}$ of ~2% and relatively lower RMSE values too. ERA5 showed consistently better performance in terms of RMSE whilst WRF-Chem d01 and d02 showed better $\overline{\Delta}$ performance in BL and UTLS respectively.

Underestimated RH variability in UTLS as well as the small and consistent negative $\overline{\Delta}$ shown in Figure 6 are mostly driven

by $\overline{\Delta}$ in the upper layer of the troposphere as Figure 5 shows variations above the tropopause are very small. All 3 instruments show a small positive $\overline{\Delta}$ in BL. In FT, ATR42 and weather balloons agree overall on a small negative $\overline{\Delta}$, despite disagreement from 800 to 600 hPa for ERA5. WRF performed better than ERA5 for RH especially when looking at d02 performance in the boundary layer.

### 3.4 Conclusions on weather data comparison

The good performance of both ERA5 and WRF in terms of wind speed and direction is not surprising as they are widely used and well validated models. ERA5 speed scores were better than both WRF d01 and d02, and direction scores were about equivalent. The fact that ERA5 is a reanalysis product could explain this, as it benefits from data assimilation unlike WRF. WRF could be expected to perform better than ERA5 in the boundary layer, given its fine resolution and use of an advanced PBL scheme to model turbulence. More precisely, $\overline{\Delta}$ should get better with higher resolution, however noise related statistics

could be expected to get worse as higher resolution implies more potential noise. We indeed find lower $\overline{\Delta}$ in both wind speed and direction for WRF over ERA5 in BL. However, performance did not improve significantly between d01 and d02, d01 even slightly outperforming d02 in multiple metrics for both speed and direction. Statistics other than $\overline{\Delta}$ are all influenced by noise even though RMSE and $\sigma_N$ do not depend solely on it. Thus we use CRMSE, represented by radial distance from the reference point in Taylor diagrams, to assess model noise performance. We indeed find that WRF d01 and d02 have slightly

higher CRMSE than ERA5 in BL & FT for both speed and direction which confirmed the expected relative behaviour of these models. WRF outputs can be improved by nudging, which consists in adjusting model estimates using observations or reanalysis products, to help regional simulations fit observations better (Bullock et al., 2014).

Assessment of temperature was also characterised by an overall very good performance from all simulations ($\overline{\Delta}$ < 1K in all layers). Temperatures from weather balloons appear to be slightly biased (by about 2 K) in BL. This could be due to a

lack of corrections of temperatures measured in the boundary layer by the M20. Further checks did not find any correlation between wind speeds and $\overline{\Delta}T$ as measured by the instrument, so no physical disturbance appeared to have been interferring with measurements. This was unexpected as calibration was performed prior to balloon release on the ground, in that surface layer. It is worth noting that consistent $\overline{\Delta}$>0 was only found in some flights (002, 003, 004 on 21/08 and 22/08) that had $\overline{\Delta}$ > 1K in BL, the other half of the flights not showing this characteristic. Investigating those particular flights in more detail

appears necessary to understand the origin of our findings.





Overall, WRF simulations were close to both ERA5 and MAGIC2021 data in terms of performance gives confidence in the model's ability to simulate the atmosphere in the region of interest.

## 4 Assessment of CH$_4$ simulations

### 4.1 Comparison between modelled and observed CH$_4$ profiles

Figure 7 shows the average difference between modelled CH$_4$ mixing ratios and MAGIC2021 measurements, noted $\overline{\Delta}$CH$_4$. In this figure we chose to only show comparison results of the PLS Surf b inversion from the provided 6-ensemble inversion optimised products as it performed better than other simulations in all layers (as shown in Figure 8). BL was characterised by a $\overline{\Delta} \leq 0$ for all global models and a positive $\overline{\Delta}$ of 20-50 ppb for WRF-Chem d02 and 20-100 ppb for d01. PLS Surf b notably showed a $\overline{\Delta} \sim 0$ when compared with AirCore data but a significant underestimation of both Cessna and ATR42 measurements.

In FT, CAMS *hlkx* consistently showed a negative $\overline{\Delta}$ of 25-50 ppb against ATR42 and AirCore data, whilst PLS Surf b showed $\overline{\Delta} \sim 0$ when compared with AirCore data, but a more significant negative bias than all other models when compared with ATR42 data. CAMS v21r1 still showed a consistent $\overline{\Delta} \sim 0$. For regional products, $\overline{\Delta}$ decreased significantly from the start of FT (P=800 hPa) up to 300 hPa, also reducing the gap between d01 and d02, so performance in that layer was close to that of CAMS v21r1 and PLS Surf b. In UTLS, CAMS *hlkx* $\overline{\Delta}$ went from the negative values of FT to a strong positive $\overline{\Delta}$ of >200

ppb at its highest, with a consistent gradient from P=300 to P=45 hPa. An overall positive $\overline{\Delta}$ was also observed with PLS Surf b, CAMS v21r1 and WRF-Chem in UTLS although the $\overline{\Delta}$ profile structure was more complex. Indeed, these 4 profiles showed a first increase in $\overline{\Delta}$ at the very start of UTLS (P=300 hPa) up to $\sim$250 hPa where $\overline{\Delta}$ hit a maximum value of 50-100 ppb depending on the model. $\overline{\Delta}$ then decreases to -50-0 ppb from P$\sim$175 hPa to P$\sim$100 hPa, before increasing again to $\sim$100-200 ppb at P<100 hPa.

After being mainly influenced by surface emissions in BL, WRF-Chem mixing ratios are mainly influenced by long range transport of bounday conditions in FT and UTLS. As these were provided by CAMS v21r1 in our simulations, both d01 and d02 profiles become similar to CAMS v21r1 at these levels. A deviation from this behaviour is however seen in UTLS, above the first peak in $\overline{\Delta}$ at P$\sim$300 hPa. This is likely due to transport differences between WRF-Chem and TM5 (transport model used for the CAMS v21r1 product) in the region. Among inversion optimised simulations, CAMS v21r1 appears as the better

product in terms of $\overline{\Delta}$, showing similar performance to PLS Surf b at the bottom of the atmosphere, but better performance in top layers.

Figure 8 shows that highest correlations between modelled and measured CH$_4$ concentrations were found in UTLS, with consistent $\rho$ values of 0.96-0.98 for all global models and $\sim$0.85 for both WRF-Chem domains. Among global models in UTLS, CAMS v21r1 performed best in terms of $\sigma_N$. Regional WRF-Chem products ranked higher than global products in

that metric when considering UTLS alone. In that layer, both regional products slightly overestimated $\sigma_N$ whilst global models showed a tendency to underestimate it. In BL and FT $\rho$ performance was much worse for all models, as only CAMS *hlkx* had $\rho > 0.7$ in FT. Most models showed similar $\rho$ performance in BL and FT. Some models performed well in terms of $\sigma_N$ in BL and FT, notably WRF-Chem d02 and CAMS *hlkx* in FT and PLS in BL. BL WRF-Chem and CAMS *hlkx* CH$_4$ showed a



**Figure 7.** Methane intercomparison between MAGIC2021 data and chemistry-transport models. Profiles are computed using the full AirCore dataset and profile sections of ATR42 and Cessna flights. **Top:** Mean CH4 profiles accross all flights from each aircraft (dotted grey line) plotted with modelled CH4 interpolated on aircraft trajectories: Cessna (left), ATR42 (centre) and weather balloon (right). **Middle:** Vertical profile of mean CH4 difference between measurements and models for each aircraft: Cessna (left), ATR42 (centre) and weather balloon (right). $\Delta CH_4 = \overline{CH_4}^{mod} - \overline{CH_4}^{obs}|_L$ for atmospheric level L. **Bottom:** Sections of mean CH4 difference vertical profiles corresponding to the 3 analysis levels: BL (left), FT (centre) and UTLS (right), where Cessna data is shown in dashed lines, ATR42 data in dotted lines and AirCore data in solid lines. Shaded areas represent the $1\sigma$ deviation from the mean CH4 or CH4 difference profiles.



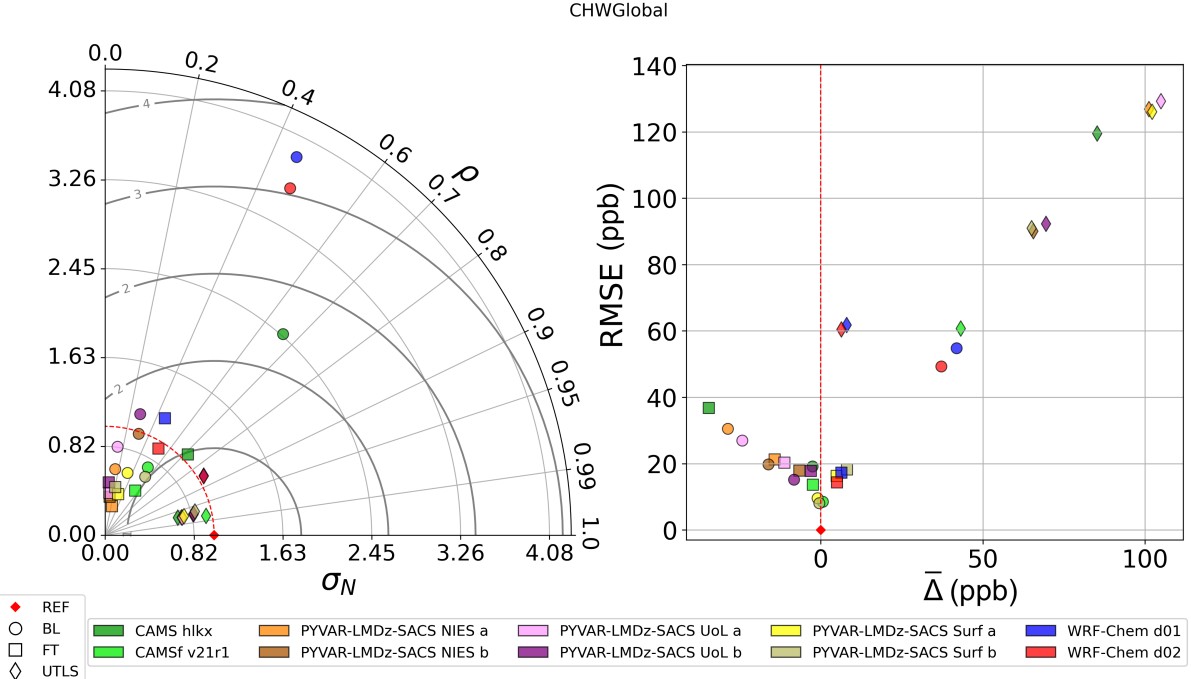

**Figure 8. Left:** Taylor diagram for CH$_4$ comparisons between MAGIC2021 observations and ERA5 model. The radial axis represents the normalised standard deviation of the modelled CH4. The angular axis represents the correlation between modelled and observed CH$_4$. The centred RMSE is represented by the radial distance from the reference point. **Right:** RMSE against $\overline{\Delta}$CH$_4$ computed from MAGIC2021 observations and modelled CH$_4$.

strong overestimation of $\sigma$ whilst PLS showed a performance of similar quality in both BL and UTLS with 0.6<$\sigma_N$<1. This

model ensemble was characterised by a relatively strong underestimate of $\sigma$ in FT where 0.25<$\sigma_N$<0.5 for all 6 products. In terms of RMSE and $\overline{\Delta}$, it was not surprising to find models worst performance in UTLS considering the comparison shown in Figure 7. Among global models, CAMS v21r1 had the best performance in terms of both RMSE and $\overline{\Delta}$. PLS ensemble showed a disparity between classical (a) and standard (b) physics schemes in that layer, with the latter performing consistently better than the former with respect to both metrics. WRF-Chem d01 and d02 outperformed all other model products in UTLS, having

RMSE values similar to CAMS v21r1 but a lower $\overline{\Delta}$. CAMS *hlkx* showed a similar level of performance as PLS products with classical physics in that layer. Models were nearly split in half in terms of BL vs FT performance, with worse RMSE and $\overline{\Delta}$ in BL compared to FT for PLS NIES a,b UoL a as well as WRF-Chem d01 and d02, whilst CAMS *hlkx*, CAMS v21r1, PLS Surf a & b performed better in BL. The exception was PLS UoL b which showed a better RMSE performance in BL than in FT but a worse $\overline{\Delta}$ in BL. BL performance was particularly good for both PLS Surf products and CAMS v21r1, recording the lowest

RMSE and $\overline{\Delta}$ values. WRF-Chem d01 and d02 showed worse performance in BL compared to all global simulation products but were comparable in FT. WRF-Chem d02 showed better performance than d01 in all layers.





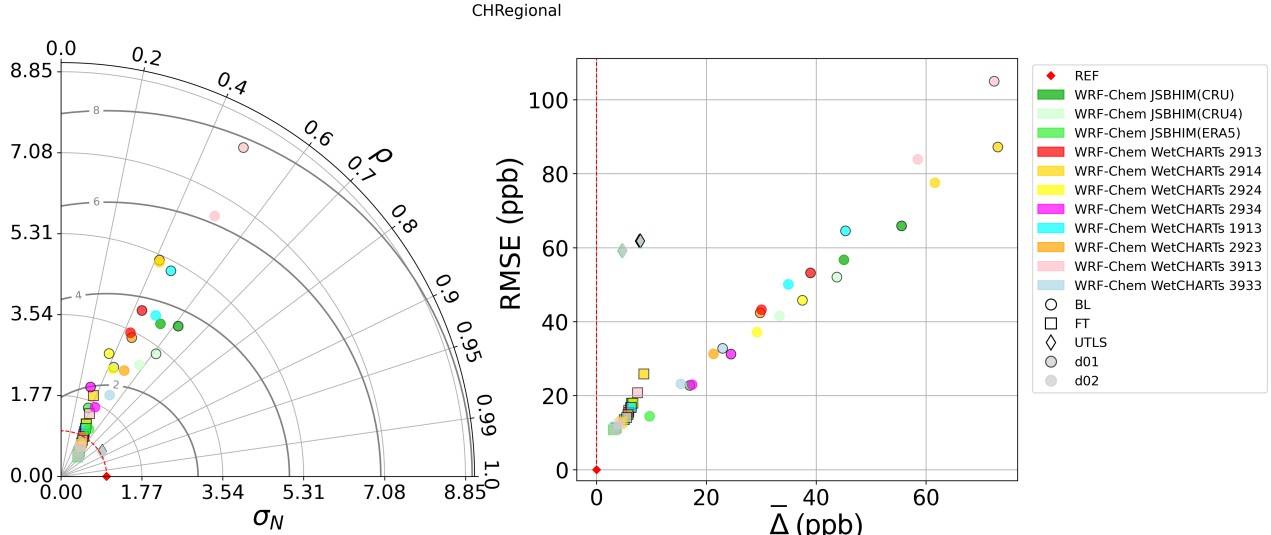

**Figure 9.** Statistical assessment of WRF-Chem simulations against MAGIC2021 measurements **Left:** Taylor diagram for $CH_4$ comparison between MAGIC2021 observations and WRF-Chem. The radial axis represents the normalised standard deviation of the modelled $CH_4$ , whilst angular position represents correlation between modelled and observed $CH_4$ . Centred RMSE is represented by the radial distance from the reference point. **Right:** RMSE against $\overline{\Delta}CH_4$ computed from MAGIC2021 observations and modelled $CH_4$.

## 4.2 Discussion of $CH_4$ comparisons

Five of the eight global model products showed a negative $\overline{\Delta}$ in the boundary layer with relatively poor correlation and an underestimate of variability, contrary to WRF-Chem mixing ratios which showed both a positive $\overline{\Delta}$ and an overestimate of $\sigma_N$.

This is consistent with an understimate/overstimate of surface emissions as weak sources would both lead to a negative $\overline{\Delta}$ and a decrease in variability, whilst overestimated surface emissions would lead to both a positive $\overline{\Delta}$ and an overstimated variability of boundary layer mixing ratios. This could also be explained by an underestimation of BL height by WRF, which would give higher $CH_4$ concentrations. WRF d01 and d02 results presented in Figures 7 and 8 are an average over eleven different products for each of d01 and d02. As such, some of the products did perform better than others within the ensemble. More particularly,

$CH_4$ mixing ratios from the WRF-Chem simulation driven by JSB-HIM(ERA5) emissions (shown on Figure 7 in green) were closest to boundary layer measured mixing ratios over the duration of MAGIC2021, with performance similar to CAMS v21r1. This product had the lowest wetland emissions out of all our inventories over MAGIC2021 duration and area. To investigate results from regional simulations in more depth, we show results from individual WRF-Chem products in Figure 9.

     Mixing ratios in the BL are positevely biased for all products, thus we deduce that inventories overestimated the magnitude

of wetland emissions (which also leads to overestimating flux variability). WRF-Chem d02 products performed systematically better than d01 in all 3 layers over 3 of our 4 statistical metrics ($\rho$ performance was inventory dependent and very close between d01 and d02). These results first showed that low emissions are needed to match observations when looking at averages over



the whole MAGIC2021 dataset. Wetland methane releases are typically not homogeneously distributed and continuous (Rinne et al., 2018; Waletzko and Mitsch, 2014), and causes for differing fluxes are multiple, so hard to fully encompass in inventories.

This is reinforced by the fact the not only WetCHARTs monthly averaged emissions lead to such overstimates, but also JSB-HIM products which have a daily time resolution as well as a higher spatial resolution and more complex underlying emission processes. Without representing this complexity, emissions overestimate can be happen by not accounting for natural emissions variability. Higher temporal and spatial resolutions combined with better knowledge of driving processes and local ecosystems therefore seem needed to accurately represent wetland emissions in inventories. The resolution of global models does not

allow to represent $CH_4$ mixing ratio variations at the scale of in-situ measurements. This motivates the use of higher resolution models such as WRF-Chem for tracking natural wetland emissions, as their advantage over global models is their ablility to simulate high resolution transport in the boundary layer which could help reproduce high resolution signals. Moreover, better resolution was shown to also improve simulation performance especially and crucially in the BL so high resolution models seem needed to have the most accurate results. Other modelling techniques, such as Lagrangian particle dispersion models ,

could also be used to study these emissions.

A tropospheric negative $\overline{\Delta}$ between simulations similar to CAMS *hlkx* (C-IFS forecast) and AirCore data was also found by Membrive et al. (2017) on a high resolution profile measured in Canada during the StratoScience campaign (CNES - August 2014) during which a high resolution AirCore (AirCore-HR) was deployed on a stratospheric balloon flight near Timmins, ON. (48.6°N). This study compares well with ours because similar $CH_4$ sources can be found near both locations, and data

was also collected in August. Membrive et al. (2017) found $\overline{\Delta}$ = -24 ppb when comparing AirCore measurements to the C-IFS forecast. We find an overall tropospheric $\overline{\Delta}$ of -14.7 ± 16.6 ppb when comparing MAGIC2021 versus CAMS *hlkx*, which is a comparable result. Further conclusions cannot be drawn from comparing these two studies alone, but this feature is also consistently found when comparing AirCore profiles from the AirCore-Fr network with CAMS forecast and analysis products (Crevoiser et al., in prep., Koffi and Bergamaschi (2018)). This suggests the presence of a systematic bias in CAMS forecast

and analysis products.

Whilst a significant tropospheric $\overline{\Delta}$ was only found between CAMS *hlkx* and MAGIC2021 measurements, UTLS analysis highlighted the presence of a strong positive $\overline{\Delta}$ for all models (cf. Figure 7). Figure 10 shows all MAGIC2021 AirCore profiles plotted along with mean and spread of corresponding interpolated model profiles. Measured $CH_4$ profiles show three distinct phases in UTLS. The bottom of the layer is characterised by a first strong gradient, typically from 400-300 to 200 hPa, which

takes mixing ratios from their tropospheric average of ∼1950 ppb to about 1810 ppb. Concentrations then remain stable for 100 hPa or less before starting a sharp decrease again in the last layer, between 200 and 100 hPa. This overall structure is in reality more complex when looking at individual profiles and highlights the stratification of the atmosphere at these altitudes. Modelling this stratification and chemical content of separate layers accurately is crucial to match observed $CH_4$ concentrations at this altitude because they can vary strongly over a short vertical distance depending on chemical content of air masses through

which measurements are made. CAMS *hlkx* has three to four times as many vertical levels as other products that are compared to MAGIC2021 observations. Within UTLS (P<300 hPa), and especially at its bottom, it makes an important difference in terms of structure complexity of profiles. As such, CAMS *hlkx* $\overline{\Delta}$ profile does not show the positive $\overline{\Delta}$ peak of 50-100 ppb



**Figure 10.** Modelled and MAGIC2021 AirCore CH$_4$ profiles plotted against pressure. Displayed modelled profiles are mean of all interpolated profiles for each model and the coloured area represents the spread between all profiles. From the PLS ensemble, only the best performing product (Surf b) is shown.





between 300 and 200 hPa as inversion-optimised models (PLS, CAMS v21r1) do. This is because it can capture a more realistic vertical structure of $CH_4$ depletion near the tropopause, as shown in Figure 10 (it is also helped by its initial negative $\overline{\Delta}$ in the troposphere). A possible way to improve the performance of chemistry-transport models at UTLS levels would be to couple them with models that focus on stratospheric chemistry, such as REPROBUS (Lefèvre et al., 1994, 1998; Jourdain et al., 2008), which implement stratospheric chemistry in more detail, notably taking into account more $CH_4$ sink molecules, thus potentially preventing $CH_4$ overestimates. Verma et al. (2017) and Membrive et al. (2017) attribute UTLS $\overline{\Delta}$>0 to an understimation of the $CH_4$ stratospheric gradient. If that were the case, a continuously growing positive bias in the stratosphere would be observed between modelled and observed concentrations. This works for CAMS *hlkx* from P=300 to P=45 hPa as seen on the bottom right panel of Figure 7, which is a similar product to the one assessed in Verma et al. (2017) and Membrive et al. (2017). However, our results show that below 45 hPa, $CH_4$ $\overline{\Delta}$ actually decreases between CAMS *hlkx* and observations. In fact, $\overline{\Delta}$ decreases below ~50 hPa for all global models. For inversion-optimised models, these variations form a second $\overline{\Delta}$ peak of 100-200 ppb between 80 and 30 hPa, and $\overline{\Delta}$ stops increasing at lower altitudes for CAMS v21r1. These results contrast with a simple continuously growing $\overline{\Delta}$, suggesting that other factors may also induce a stratospheric $\overline{\Delta}$ in $CH_4$. Figure 10, showed that $CH_4$ values from AirCores started to decrease strongly at lower altitudes than in models, suggesting that the influence of chemistry near the tropopause is vertically delayed for all models. As such, we hypothesise that both problems (weak gradient and delayed $CH_4$ decline) influence CAMS *hlkx* and PLS products. This delayed decline could be explained by both weak chemistry and a lack of stratosphere-troposphere interaction. CAMSv21r1 and PLS show a more realistic $CH_4$ gradient, but their lower vertical resolution does not allow to resolve the stratification of the atmosphere at these altitudes, which causes observed biases as these product. It is worth noting that a difference in tropopause height between models and observations does not influence the results, as confirmed by temperature profiles (Figure 4).

### 4.3 Conclusions on $CH_4$ comparisons

Our model performance intercomparison highlights important differences between MAGIC2021 observations and modelled $CH_4$ mixing ratios, especially in UTLS levels where all models overestimate atmospheric methane content. CAMS *hlkx* analysis showed highest $\overline{\Delta}$ of all models in the UTLS and also suffered from consistent underestimation of atmospheric methane content in the FT. Inversion-optimised products showed better perfomance at every levels than CAMS *hlkx*. However, CAMS *hlkx* denser vertical grid at high altitude proved to be a certain advantage to better resolve tropopause chemistry. Among inversion optimised global chemistry-transport models, CAMS v21r1 showed the best performance in terms of $\overline{\Delta}$. Standard physics and surface observational constraints were found to be the best combination within the 6 PLS ensemble inversions, this version (Surf b) showing a similar level of performance as CAMS v21r1. Regional simulations were characterised by a strong overestimation of BL $CH_4$ atmospheric content, which was not found in global simulations. This overestimation shows that inventory wetland emissions used in our simulations were too high on average.



# 5  Conclusions

ERA5 reanalysis and WRF simulations were assessed using meteorological data from MAGIC2021. Methane in-situ measurements from MAGIC2021 were also exploited to assess atmospheric composition models: the analysis product CAMS *hlkx*, the inversion-optimised product CAMS v21r1, six PLS ensemble inversions and WRF-Chem simulations. Over the six days of MAGIC2021, meteorological data from ERA5 showed better agreement with observations than WRF on average, due to both data assimilation and lower resolution that enhance performance in such an exercise. WRF performance was however

very close for all physical quantities assessed, which gives us confidence in its ability to simulate regional atmospheric physics for MAGIC2021. Among global simulations, inversion-optimised simulations of $CH_4$ concentrations performed best, especially close to the surface. CAMS v21r1 showed slightly better performance than PLS ensemble inversions. A detailed analysis of regional simulations with WRF-Chem was performed, revealing perfomance disparities among $CH_4$ products. Overall we observed only positive biases in the boundary layer, indicating a tendency to overestimate emissions by wetland emissions

models. $CH_4$ profiles were also characterised by performance discrepancies near the tropopause, where $CH_4$ content is depleted by reactions with OH radicals and can also be affected by stratospheric intrusions. All models showed a delayed vertical gradient of $CH_4$ mixing ratios near the tropopause, leading to a positive bias in the stratosphere. Comparisons with CAMS *hlkx* showed that high vertical resolution allows to better capture vertical structure of $CH_4$ profiles in the stratosphere, with a large overestimate still. These results call for more work dedicated to improve the chemistry of models in the UTLS, which

could be done by separate stratosphere models, specialised in the task. Finally, we aknowledge that the MAGIC2021 dataset is limited in both spatial and temporal extents, limiting its ability to fully assess models. However, the results presented here represent a rare opportunity to assess the performance of models against a large, high resolution dataset, over an area where few measurements are usually taken. This highlights the need for more frequent extended campaigns at high latitudes to fully characterise local processes and extend our performance assessment of global and regional models.

*Data availability.*  Data from MAGIC2021 is available on the French national center for Atmospheric data and services AERIS catalogue, and from the HALO (Re3data.Org, 2016) database

*Author contributions.*  - FL & CC designed the study

- FL performed the comparisons between MAGIC2021 data and model outputs

- ERA5, CAMS *hlkx* and CAMS v21r1 data was retrieved by JP

- WRF-Chem simulations were designed and run by FL with support from TL and CA

- MS and XL provided the PYVAR-LMDz-SACS ensemble data

- Weather balloon data (AirCore + M20) was produced by JP, AG and TP

- ATR42 $CH_4$ data was treated by AG and FL

- Cessna data were acquired and provided by AF, K-DG, AR as well as Heidi Huntrieser, Vladyslav Nenakhov and Magdalena Pühl





- Cessna meteorological data was produced by Vladyslav Nenakhov

  - Cessna $CH_4$ data was produced by AF

  - Analysis of results was done by FL with support from all co-authors

*Competing interests.* The authors declare that they have no conflict of interest.

*Acknowledgements.* We thank CNRS MITI for funding this research as well as CNRS, CNES and ESA for funding MAGIC2021. This study
also benefited from the IPSL Data and Computing Center ESPRI which is supported by CNRS, SU, CNES and Ecole Polytechnique, as
  well as the ROMEO HPC center at the University of Reims Champagne-Ardenne. T. Lauvaux was supported by the French fellowship Make
  Our Planet Great Again (CIUDAD, CNRS), the French Ministry of Research (Junior Chair professor CASAL) and European Space Agency
  (project MethaneWatch).

  *Carbon footprint.* The full carbon footprint of the MAGIC2021 campaign is still being estimated. For this study, we compute
an approximate value based on the highest emitters: aircraft flights. The Cessna from DLR has a 675HP turbine engine and flew
  for 27h22min, which according to Labos1point5 data equates to an emissions of $13 \pm 1$ tCO$_2$. The ATR42 from SAFIRE has a
  3800HP turbine engine and flew for 25h22min, which equates to emissions of $32 \pm 2$ tCO$_2$. Therefore the total carbon footprint
  of aircraft flights associated to this paper is $45 \pm 2$ tCO$_2$. The balloon's carbon footprint is more complicated to estimate. Most
  recoveries were performed using a helicopter for which engine and flight time data were not part of the MAGIC2021 dataset,
resulting in a lack of information. Additionally, the helium used to inflate campaign balloons is a potent greenhouse gas that
  is released in the high troposphere/lower stratosphere everytime a balloon is used. Working out the full carbon footprint of
  radiosoundings therefore requires converting released helium to CO$_2$ equivalent which has not yet been done for MAGIC2021.
  The carbon footprint of campaign measurements involved in the study presented here is therefore not complete, and probably
  totals to more than 50 tCO$_2$. The campaign as a whole will have a higher carbon footprint still, as it includes the footprint of
meals provided during the campaign, travels to Kiruna for every team, additional airborne measurements that were not used in
  this paper, as well as tools, clothes and instruments that were bought especially for MAGIC2021 . Also neglected here is the
  footprint of the data analysis and model simulations post-campaign, which are run using high performance computing facilities.
  Carbon footprint numbers given here are therefore neither representative of the whole campaign nor of the data analysis and
  modelling footprint, so it should be considered as a lower bound for the footprint of this paper only.



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
