# Peer review of "Evaluating Weather and Chemical Transport Models at High Latitudes using MAGIC2021 Airborne Measurements"

_EGUsphere, 2024_

## Author Response (AR1)

Dear Mr. Custodio,

Thank you for reviewing our manuscript, "Evaluating Weather and Chemical Transport Models at High Latitudes using MAGIC2021 Airborne Measurements," for publication in Atmospheric Measurements Techniques. The insightful comments and valuable suggestions have been instrumental in improving the quality of our manuscript.

Below, we provide a point-by-point response to your comments and concerns, with our responses indicated in blue. All page numbers refer to the revised manuscript file.

**1. Major Concerns**

**Clarity and Presentation**

The manuscript is difficult to follow due to unclear wording, undued wording, and overly dense descriptions. Some key points are buried in the text, making it challenging for readers to extract the central findings and their implications. Additionally, the plots are mazy and visually overwhelming, detracting from their effectiveness in conveying the results.

> The *Results & Discussion* sections (3&4) have been rewritten entirely with emphasis on clarity and bringing out the main results. Summary tables have been added to improve the understanding of the results and avoid getting lost in too much detail in the text.

**Metrics and Model Performance Assessment**

While the study evaluates model performance, the choice of metrics is not optimal. The authors should consider employing more comprehensive and widely accepted set of statistical metrics for model evaluation. Correlation Coefficients and Root Mean Square Error are good; however, I would recommend bias.

> Bias was actually employed but improper wording led to confusion. The bias calculation was done according to Willmott (1982). This has been added l. 271, and the word "bias" was used instead of our previous unclear wording.

Additionally, the manuscript could discus the implications of the metrics used. For example, while some metrics may show agreement, others may reveal discrepancies, which are worth exploring.

> The new summary tables show more clearly these different performances between models and paragraphs were added to discuss those in more detail in the *Results & Discussion* section (l. 382 to 387 and l. 517 to 523)

**Figures**

The figures and tables are a central issue. While they contain a wealth of information, they are too crowded and difficult to interpret. Each figure should serve a clear purpose and convey specific insights. To improve:

*Use clean background, simplify the layout and make sure that the data are visible.*

Use color schemes that are easy to distinguish, particularly for readers with color vision deficiencies.

Add concise and informative captions that explain the key takeaways from each figure.

Our new figure template uses a white background and colour maps specifically designed for colour deficiencies (provided by the cmcrameri matplotlib package). We also acknowledge that we relied too much on colours to distinguish between the different plots and have implemented, where possible, different approaches based symbols. Figure captions for all figures except Fig. 1 and Fig. 9have also been reviewed for clarity.

**2. Others Comments**

The manuscript's Figure 1 is confusing and requires clarification:

Scope of Flights: Does Figure 1 intend to show the entire MAGIC2021 campaign or only flights over Kiruna? The caption does not make this clear.

> The figure intends to show the flights from the MAGIC2021 campaign that were used for this paper. All of these were over the region of Kiruna, except for one which targeted oil platforms on the Norwegian coast. Clarifications have been added in the caption of the figure.

Flight over Norway: Why the flight over Norway is not included in the figure? Is it not part of the MAGIC2021 campaign?

> The flight over Norwegian oil platforms is indeed part of MAGIC2021. We initially chose not to show it on the map in order to distinguish in detail all the other flights. However we agree that this choice makes the map unclear so all the flights used in the study are now shown on the map

Unexplained Elements: The color blocks in the middle of Figure 1 are not explained in the caption or the text. This information seems to appear "out of the blue," and the figure lacks sufficient annotation to guide the reader. Please ensure that every feature in the figure is fully explained in the caption and supported by the text.

- > Colours of the figure have been modified in order to better distinguish each category. Detail on the land cover dataset can be found in the paper whose reference is now provided in the caption text.
- > New caption for Figure 1:

Location (left) and date and time (right) of MAGIC2021 measurements used in this study. Map background shows land use adapted from the Corine2018 dataset (European Environment Agency, 2019). Individual land cover types from the Corine2018 dataset are grouped in broader categories to ease map interpretation. Notably, wetlands include inland marshes, peat bogs, salt marshes, salines, intertidal flats, coastal lagoons and estuaries, i.e. both freshwater and saltwater wetlands.

The introduction of AirCore measurements is presented in a very shallow manner. While AirCore is a critical part of the study, its role and methodology are not sufficiently explained. Readers who are unfamiliar with AirCore technology will grasp it.

> Description of the instrument has been expanded. The new expanded section can be found from l. 98 to l. 109.

The division of the atmosphere into three layers based on pressure ranges—P > 800 hPa, 300 < P < 800 hPa, and P < 300 hPa—is arbitrary and does not align with commonly accepted atmospheric definitions. The chosen pressure thresholds do not accurately correspond to the planetary boundary layer (PBL), free troposphere (FT), or lower stratosphere (LS). A more scientifically sound approach would involve:

Using PBL height (PBLH) to define the boundary layer.

Defining the tropopause to separate the troposphere from the stratosphere.

This approach would ensure that the results are more meaningful and interpretable, especially for discussions of  $CH_4$  transport dynamics across these atmospheric layer

> We agree that this arbitrary definition was a blind spot of the study. The three analysis levels were redefined according to this comment. The first level (BL) was redefined according to boundary layer height data from ERA5. More precisely, BL height was interpolated from ERA5 on the aircraft and balloon trajectories, then compared to flight height and flagged as 'in BL' if the flight altitude was below the BL height interpolated from ERA5. Tropopause height was determined from AirCore measurements using the

Cold Point Tropopause method (Section 2.3.2 starting l. 267 was updated). The same method was used on ERA5 data to assess its accuracy against balloon observations. As temperatures are in good agreement between ERA5 and MAGIC2021 at the tropopause, the tropopause height derived from both datasets are very close.

The table captions should be placed at the top of the tables, following standard formatting conventions. Additionally, the table labels should succinctly describe the contents of the table. For instance, it does not make sense to include information about what is not in the table. Ensure that the captions are clear and concise, helping the reader to quickly understand the data presented.

> Tables' caption placement was changed according to this comment. Superfluous information has been removed from table captions where found.

The manuscript refers to "four statistic," which is an unclear and incorrect phrasing. Likely, the authors mean "four metrics used to evaluate model performance." The use of appropriate and precise terminology is critical for clarity. This error is indicative of broader language issues in the subsection "Statistics," which should be rewritten to ensure proper English usage and a professional tone.

> The section has been rewritten according to the reviewer's comment. Additionally, the word "statistic(s)" has been replaced by "metric(s)" in other sections too.

The caption for Figure 2 is insufficient to help readers understand the plot. Captions should summarize the key information conveyed in the figure and provide any necessary context for interpretation. In its current state, the caption leaves too much ambiguity and fails to assist the reader in navigating the content.

**> New Figure 2 caption:**

Wind rose plots for MAGIC2021 observations as well as ERA5 and WRF simulations. The radial axis gives the proportion (in %) of wind coming from a given direction given by the angular axis. Coloured bins represent the share of speed ranges shown in the legend associated with each direction. Rows correspond to data products MAGIC2021 observations, ERA5, WRF d01 and WRF d02. Columns correspond to atmospheric layers: Boundary Layer (BL), Free Troposphere (FT) and Lower Stratosphere (LS), described in Section 2.3.2.

The manuscript's discussion of wind fields is constrained solely to advection (horizontal transport), which provides an incomplete picture. The vertical component of wind, which is critical for transport processes and atmospheric mixing, is entirely missing. Vertical transport are among the most significant challenges in atmospheric modeling. Without addressing these, the discussion remains superficial. The authors could evaluate turbulence representation and vertical wind components in the models, as these are critical to understanding transport processes.

> This paper aims at comparing campaign data to simulation data. Unfortunately vertical wind data quality was inconsistent in the MAGIC2021 dataset, particularly with the weather balloon instruments. Therefore the decision was taken to focus on horizontal winds for which data could be compared to models.

Figure 5 is visually confusing and "weird" in its current presentation. The layout, formatting, and choice of visualization make it difficult to follow and interpret. Clearer design and simpler representations would greatly enhance the reader's understanding of this figure. Ensure that key messages are apparent and not lost in the visual clutter.

The content of subsection 3.3 is difficult to follow due to poor organization and unclear visualizations. The comparisons presented in this section lack coherence in terms of visual representation, metrics used, and overall wording. It is essential to streamline the presentation of comparisons to make them more reader-friendly and effective.

> This section was fully rewritten, with the addition of a summary table at the end of section 3. in order to help the reader comparing the model results.

The comparison of meteorological data between models and observations is superficial, merely reporting which model or dataset is closer to observations. This approach fails to provide meaningful insights or a deeper understanding of model inter-comparisons. Readers expect a more insightful analysis of model performance, including:

*Identifying potential reasons for discrepancies.*

Explaining how differences in parametrizations or data assimilation processes contribute to observed biases or differences.

Suggesting ways to improve model representation of meteorological processes.

The manuscript must go beyond simply reporting agreement or disagreement to provide a more nuanced and insightful evaluation.

> Section 3.4 was largely expanded. We include a discussion on the discrepancy between WRF and ERA5 performance. We also give a possible explanation as to why the increase in domain resolution between WRF d01 and d02 simulation does not directly equate to improvements in all statistical categories, based on Mass et al., 2002; Gómez-Navarro et al., 2015. We suggest implementing data assimilation in WRF to test its impact on model performance compared to high resolution observations.

The vertical profiles presented in the manuscript are overly complicated and lack clarity. The plots are "mazy," and the text does not provide sufficient guidance to help the reader interpret them. The analysis of vertical profiles should do more than report which model performs better in specific atmospheric regions (which, as noted above, were not properly defined). A thorough discussion of the physical processes contributing to vertical variations in CH4 and meteorological variables would enrich the article.

The conclusion that all models overestimate  $CH_4$  at the upper troposphere-lower stratosphere (UTLS) boundary is interesting but could be influenced by the interpolation method used for data colocation. In addition:

TM3 does not have the resolution to accurately resolve the tropopause.

While IFS has more vertical level, it still struggles with tropopause representation.

The lack of proper selection for the lower-most stratosphere in this study further compounds this issue. A more refined methodology is required to draw robust conclusions about model biases in the UTLS region.

> Additionally to the redefinition of analysis layers, we added more discussion points about the stratospheric bias which we hope give insights about where it could come from and what could be done to investigate the issue further (l. 510 to 554).

The association of the overall positive  $CH_4$  bias in the boundary layer to wetland emissions is an important finding. However, this conclusion seems premature without further testing. A sensitivity test maybe could strengthen this claim and ensure that this conclusion is robust.

> Information has been added in the Methods section about the emission products used for WRF-Chem simulations (Table 1). This information helps to link overestimates seen in the lower troposphere to the emissions products. We also added literature references and other analysis elements to toughen that claim (l. 481 to 489).

The spatial and temporal limitations of this model evaluation could be addressed by incorporating data from the CoMet 2.0 campaign over Canada in the summer of 2022. While the MAGIC2021 campaign provides valuable observations, supplementing this with additional datasets could offer a more comprehensive evaluation of model performance.

> This comment has been added in the *Conclusion* section.

**References**

Willmott, C. J. (1982). Some comments on the evaluation of model performance. Bulletin of the American Meteorological Society, 63(11), 1309-1313. DOI

Mass, C. F., Ovens, D., Westrick, K., and Colle, B. A.: Does Increasing Horizontal Resolution Produce More Skillful Forecasts?, 2002

Gómez-Navarro, J. J., Raible, C. C., and Dierer, S.: Sensitivity of the WRF Model to PBL Parametrisations and Nesting Techniques: Evaluation of Wind Storms over Complex Terrain, Geoscientific Model Development, 8, 3349–3363, DOI, 2015

Dear Sir or Madam,

Thank you for reviewing our manuscript, "Evaluating Weather and Chemical Transport Models at High Latitudes using MAGIC2021 Airborne Measurements," for publication in Atmospheric Measurements Techniques. The insightful comments and valuable suggestions have been instrumental in improving the quality of our manuscript.

Below, we provide a point-by-point response to your comments and concerns, with our responses indicated in blue. All page numbers refer to the revised manuscript file.

**1. Major Concerns**

There was no exploration of model representation of PBL height or vertical mixing. Accurate representation of these parameters and processes is a central issue in flux estimation and therefore is a bit of a gaping hole in this paper. For example, the authors conclude that some wetland models over-estimate emissions since simulated CH4 values are too high on average in the BL for many cases. How do you know that representation of BL height or vertical mixing are not playing a role in this high bias?

> BL height is now determined from interpolated ERA5 data for all comparisons and tropopause height from balloon data. This is explained in Section 2.3.2, from l. 267 to l. 271.

To determine that it methane excess in the BL was most probably due to wetland model emissions, we looked CH4 at vertical profiles. If the excess of methane in the BL were due to issues with BL height representation then there would be a negative bias (= lack of methane) in the FT layer, which we do not observe. We also added columns in Table 1 that provide information about the amplitude of wetland CH4 emissions. We note that the biases observed in WRF-Chem simulations' BL correspond to the different amplitude of the input emissions. Finally we note that our WRF-Chem set-up has been used in other studies Lauvaux et al., (2012); Lauvaux et al., (2016) without showing transport errors that would explain such biases in the BL. These elements were added in the paper l. 462 - 481.

Section 3.2, Figure 4 bottom left panel – I don't understand why models would track temperatures from aircraft well and not AirCore. I'm concerned that the model comparison has identified a significant measurement bias in the balloon-borne temperature measurements. Later in section 3.4, you say "Temperatures from weather balloons appear to be slightly biased... due to a lack of corrections..." Bad data should not be retained for model evaluation, even if it's discovered after the fact. For validation of all observational variables analyzed, were measurements from separate platforms/instruments inter-compared or calibrated to a common standard?

> To simplify the Results and Discussion sections (sections 3 and 4) we decided to move the comparisons between each MAGIC2021 instrument and models to the Appendix section. The new plots for Figures 5 and 7 compare the MAGIC2021 dataset as a whole against individual models. We then decided not to include in the main text this discussion about data quality.

However, to answer your question, AirCore data differs from other platforms in several ways, and the data from these other platforms agree well with models. Thus, an issue with this instrument was hypothesised instead of an actual model bias. As said in the article, further tests did not show issues with the AirCore data so more investigation needs to be done before labelling the data as "bad" or the model as biased in this particular case.

There was an inter-calibration protocol for all measurements made using the Picarro analysers, and an inter-calibration exercise between the two aircraft. This information was also added l. 86-88

Section 4.1: "PLS Surf b notably showed a  $\Delta$  ~0 when compared with AirCore data but a significant underestimation of both Cessna and ATR42 measurements." Is this finding because the CH4 data from the different platforms are biased relative to each other or because they have different processes/ecosystems in their influence regions? It would be helpful to characterize the dominate flux processes in the footprints of the observations.

> Comparisons between instruments have been moved to the appendix. However this particular issue appears to have been an artefact from our previous arbitrary layer definitions. In the new graphs shown on Figure A3, where BL height is now defined from ERA5, we don't see these differences anymore.

Section 4.2: You say some models over-estimate wetland emissions due to lack of complexity. But lack of complexity/variability is only one possible reason and no evidence is given. How do you know emissions aren't over-estimated on average?

> This comment was made in anticipation of another study that was done studying local CH4 emissions more specifically but it is beyond the scope of this paper, so that part was removed.

Section 4.3: Can you provide more information on the wetland models that overestimated methane emissions? How much higher, within the area of influence of the observations, were the emissions from the models that over-estimated observed CH4 concentrations compared to the models that got it about right?

> Information was added in Table 1, showing the comparative magnitude of input emissions. We do observe that wetland models with higher emissions produce higher mixing ratios and positive biases compared to observations in the BL.

Related to the composition- The paper is full of minor grammatical errors and awkward word choices. It's not a significant issue on an individual basis, but it made the paper laborious to read and detracted from my understanding in some cases. I have listed many minor suggested edits below.

> The Results & Discussion (3.&4.) sections have been fully rewritten with emphasis on clarity and bringing out the main results. Summary tables have been added to improve the understanding of the results and avoid getting lost in too much detail in the text.

**2. Minor concerns**

In the abstract, at this early point in the paper, the model names/acronyms (mainly: "CAMS v21r1", "PYVAR-LMDz-SACS", "CAMS hlkx") don't mean anything to the reader and no description is provided. It would be better to describe the models in mostly general terms in the abstract related to what aspects distinguish them and/or led to differences in performance.

> The abstract was rewritten according to this comment

A summary table or list of the different models tested and their important aspects/differences would be really helpful.

> Table 1 has been expanded to include more detail about emission products used in the regional simulations, notably the magnitude of each emission product. A second table has been written (Table 2) which summarises the key features of each model

Line 463: The AirCore-Fr network is not described so I don't know the space/time coverage of those observations. Are you saying that a systematic bias extends beyond the Arctic?

> Added reference for the AirCore-Fr network. Yes the tropospheric bias is found beyond the Arctic with CAMS forecasts and analyses (Koffi et al., 2018). This latter reference was added l. 499.

Line 496: "It is worth noting that a difference in tropopause height between models and observations does not influence the results, as confirmed by temperature profiles (Figure 4)." I don't understand what is meant by this statement. Please elaborate or rephrase.

> By this we mean that looking at tropopause height determined from the temperature profiles from both models and observations match. If temperature profiles match between models and observations then they should agree on the tropopause height. This was further checked when writing our new method for determining the 3 atmospheric layers BL, FT and LS (cf section 2.3.2, l. 267).

Line 521: I am unclear on what a "delayed vertical gradient" means. Too slow vertical mixing?

> This unclear wording attempted to describe graphically what was seen with methane decay at the tropopause and above, it was removed. Literature shows that too much mixing could actually drive an excess in methane in the LS. This is elaborated from l. 501 to 555, where references are cited.

Several sections have an inconsistent mix of past and present-tense from sentence to sentence. Please check to make consistent.

> This has been checked by co-authors in the new version.

Data availability: Data links/citations/DOIs are not provided making me doubt it's actual availability. Also what about the WRF data?

> MAGIC2021 data are soon to be available. WRF-Chem simulations product are heavy files which will be made available upon request.

Minor suggested edits by line number:

- 11: I don't understand what "among CH4 products" means. Among different prior flux models?
- > Wording was changed to: "Among global simulations of CH4 mixing ratios", 1.9
- 14: Suggest: Despite the [its] limited spatiotemporal scope of MAGIC2021 [coverage], we were able to identify the best performing transport models and to evaluate fluxes from different biogeochemical model parametrisations using the MAGIC2021 high-resolution dataset [, demonstrating the utility of insitu vertical profile datasets for transport and flux model evaluation].
- > Sentence was changed to: "Despite its limited spatio-temporal coverage, we were able to identify the best performing transport models and to evaluate fluxes from different biogeochemical model parameterisations using the MAGIC2021 high-resolution dataset, demonstrating the utility of in-situ vertical profile datasets for transport and flux model evaluation.", 1.14
  - 22-23: A few more citations would be good to support such broad statements about feedbacks.
- > Sentence was changed to: "The amount of greenhouse gas in the atmosphere and the meteorological conditions are essential components of the circumpolar climate system, where positive climate feedback loops are ubiquitous and disruptive (boreal fires (Zheng et al., 2023), permafrost (Miner et al., 2022; MacDougall, 2021), wetland emissions(Zhang et al., 2023), and albedo (Hall, 2004; Booth et al., 2024)).", l.21
  - 28: "tall tower" -> "surface measurement" 1.29
  - 34: For ABoVE, suggest citing: <a href="https://doi.org/10.5194/acp-22-6347-2022">https://doi.org/10.5194/acp-22-6347-2022</a> 1.36
  - 36: "(2022)" citation is incomplete and not listed  $\overline{m{V}}$
- 38: suggest: "[airborne] measurements of meteorological variables and atmospheric methane mixing ratios."

  \[
  \sqrt{}
  \]

- > "This latest project was notably involved in the CoMet 2.0 Arctic campaign set in Canada and Alaska in 2022 (DLR, 2022), and MAGIC2021, set near Kiruna, Sweden (67 °N). The study presented here focuses on MAGIC2021 (Crevoisier, 2021), which spanned from 14 to 27 August 2021 and included airborne measurements of meteorological variables and atmospheric methane mixing ratios, combined with weather data sounding.", l. 37- l. 40
  - 47: "Kiruna and it's surrounding [area] are...and tundra [ecoystems],..."
- > "Kiruna and its surrounding area are characterised by wetland landscapes that include small ponds to large lakes as well as peatland and various inundated soils found in both boreal forest and tundra ecosystems", l. 49- l. 50
  - 58: "Stohl (2004) have showed..." > "have shown" appears right, from Collins Dictionary)
  - 60: "the main" -> "an important"  $\sqrt{1.63}$
- 60: Check the accuracy of this statement: "OH is found mainly at the top of the troposphere and at the bottom of the stratosphere, where other chemical species also react with CH4..."
- 63: "gradient" -> "transport" > The sentence is about the strongly diminishing concentrations of methane with height, so it is about the gradient of mixing ratios, not about the transport of methane.

"At higher altitudes, an important driver of CH4 mixing ratios becomes methane depletion by OH radicals and other molecules (e.g. Cl, Li et al. (2018)). Their presence mostly affect methane mixing ratios in the upper troposphere and lower stratosphere, where stratification and reaction with these chemical species reduce drastically CH4 mixing ratios in the upper troposphere and above the tropopause. Uppertropospheric and lower-stratospheric CH4 mixing ratios are therefore characterised by a strong vertical gradient. Tropopause height and troposphere/stratosphere exchanges are thus key influences on CH4 mixing ratios (Xiong et al., 2013), and are also challenging to model accurately (Mateus et al., 2022)" l. 62-68

80: "ease and speed of data treatment" - What does this mean? All the data are not finalized or it was too much data to consider or ground data were not relevant to the goal of the analysis?

"Whilst this study does not make use of the full set of MAGIC2021 measurements due to data availability at the time of our analysis, it provides a solid example of such campaigns capability in terms of model validation."

Figure 1: With the colors used for land cover types, I can't tell the difference between agriculture, boreal forest, and non-forest natural land in the map figure. > Map colours have been changed to make different land cover types more distinguishable.

101: "DLR" acronym has not been defined. Also, is "(DLR)" supposed to be a citation to something? 🔽

"This study focuses on airborne data taken by CNES weather balloons as well as two aeroplanes, an ATR42 from SAFIRE and a Cessna from Deutsches Zentrum für Luft-und Raumfahrt (DLR)." 1.81-83

106: "allows to measure" -> "allowed for measurements of" \(\sqrt{}\)

118: "therefore contain" -> "include"; "samplings" -> "flights" or "soundings" 🔽

The paragraph describing balloon measurements has been rewritten, notably to include more detail about the AirCore instrument. l. 98 - 112

```
120: "was" -> "were" ✓
```

128: "the higher density of vertical levels" – compared to what?  $\sqrt{}$

This was an artefact from an older sentence. Now: "The high density of vertical levels in ERA5 from the mid-troposphere down to ground level allows for accurate comparison with the flights from MAGIC2021." 1.140

133: "To compare [model] humidity from to observations [of], that measured relative humidity (RH), to ERA5 humidity, given as specific humidity q, ERA5 data was converted..."

To compare humidity from models (given as specific humidity q in ERA5) to measured relative humidity (RH), ERA5 data was converted to RH

153: "The spatial resolution is of  $3^{\circ}\times2^{\circ}\times34$  levels and a [the] temporal resolution of [is] 6 hours.  $\boxed{\checkmark}$  1. 165-166

156: PLS - Is there a reference for this model with more detail? If not, some detail and references on inputs and setup seem to be missing. For example, there are no references given for both priors and obs.

Additional references have been added, l.168 -182

207: "Input emissions [(Table 1)] were chosen..." \( \sqrt{1.219} \)

222: "18 different flux versions are publicly available [from WetCHARTS], ..." 1.234

225: "3 versions of total wetland flux [from JSB-HIM, each] differing in their driving meteorology, were included in this study." 1.237-238

228-230: Suggest: "11 emission tracers and one boundary condition tracer were tracked in the simulation of total regional CH4 emissions. Boundary conditions were provided by..."  $\sqrt{1.240-241}$

232: "This was done by hourly adding a constant offset of 300ppb through the emission tracers domain boundaries [on an hourly basis]."  $\sqrt{1.244-245}$

*Table 2: Suggest rounding the numbers in this table.* ✓

286: "ERA5 had NNE contributions more important [indicated a larger fraction of NNE] than N winds, contrary to observations and WRF. However, it [ERA5] showed a distribution of wind speeds closer to observations than WRF, which had a more important [larger] share of low speed winds than observations. WRF winds were again very similar between the two domains. They were overestimating [WRF overestimated] the contribution from NNW, especially of low speed winds." > Sentence rewritten, as part of the full rewriting of the discussion

300: "there" -> "for the upper layer"; "better correlation" - Compared to what? > Sentence rewritten, as part of the full rewriting of the discussion

355: "instruments" -> "models"? > Sentence rewritten, as part of the full rewriting of the discussion

The rewritten horizontal wind analysis discussion paragraph can be found l. 290 - 302

366: "However, performance did not improve significantly between d01 and d02, [and] d01..." ✓ 1.379

371: "consists in" -> "involves" ✓

372: "to help regional simulations fit observations better (Bullock et al., 2014)[, but nudging was not utilized for the WRF runs analyzed here]."

V

**1. 389-391**

386: "[For the PLS model in this figure,] we chose to only show comparison results of the PLS Surf b [configuration from the available 6-member ensemble]."

V

"Only PLS Surf b results are shown from the 6 different model products, as it performed best overall (details shown in Figure 8, Table 4 and Figure A4)", l. 408-409

444: "so" -> "and" 🗸

447: "be happen" -> "occur" > Sentence removed for clarity purposes

465: "...an analysis products [for CH4 in the FT]."

485: "works" -> "is the case" ✓

*500:* "*content*" -> "*levels*" ✓

503: "chemistry" -> "structure"? > "to better resolve the structure of CH4 profiles at the tropopause"

*508:* "our" -> "those" ✓

514: "lower" -> "higher"? > Overall, ERA5 showed better results than WRF d01 which also showed better performance compared to d02 while having lower spatial resolution. The reasons as to why a lower resolution led to better results in this specific assessment are discussed in Section 3 from l. 377 to l. 393

519: ".. observed only positive [to near neutral?] biases.."

520: "...models[, at least for the limited region and timeframe captured by the observations.]"  $\boxed{V}$

> Spelling and grammar was checked and suggestions were added when applicable given that sections 3. and 4. had already been rewritten.

**References**

Koffi, E. and Bergamaschi, P.: "Evaluation of Copernicus Atmosphere Monitoring Service Methane Products.", Publications Office, LU, 2018.

Lauvaux, T., Schuh, A. E., Uliasz, M., Richardson, S., Miles, N., Andrews, A. E., Sweeney, C., Diaz, L. I., Martins, D., Shepson, P. B., and Davis, K. J.: "Constraining the CO2 Budget of the Corn Belt: Exploring

Uncertainties from the Assumptions in a Mesoscale Inverse System", *Atmospheric Chemistry and Physics*, 12, 337–354, https://doi.org/10.5194/acp-12-337-2012, 2012.

Lauvaux, T., Miles, N. L., Deng, A., Richardson, S. J., Cambaliza, M. O., Davis, K. J., Gaudet, B., Gurney, K. R., Huang, J., O'Keefe, D., Song, Y., Karion, A., Oda, T., Patarasuk, R., Razlivanov, I., Sarmiento, D., Shepson, P., Sweeney, C., Turnbull, J., and Wu, K.: "High-730 Resolution Atmospheric Inversion of Urban CO2 Emissions during the Dormant Season of the Indianapolis Flux Experiment (INFLUX)", Journal of Geophysical Research: Atmospheres, 121, 5213–5236, https://doi.org/10.1002/2015JD024473, 2016.

---

## Author Response (AR2)

**Final response to minor edits**

13: "its' is not defined. Suggest: "Despite the limited spatiotemporal coverage of the observations..."

> "Despite the limited spatio-temporal coverage of the observations, we were able to identify the best performing transport [...]" l.13-14

Figure 1: Consider using a color for the ATR42 flight track that is different than Sea and Ocean landcover type

> New ATR42 trajectory color is black

85: Hall et al. 2021 is about CO2 and this paper is about CH4 so the reference is not appropriate.

> Changed the reference to Dlugokencky et al., 2005, "Conversion of NOAA atmospheric dry air CH4 mole fractions to a gravimetrically prepared standard scale", DOI:10.1029/2005JD006035

125: The Picarro G2401 also measures CO, but the G5310 measures CO with higher precision. The 5310 also measures N2O.

> "After sampling, AirCores were retrieved and analysed on the ground, using the G2401 instrument from Picarro® (Picarro, 2008) which measured CH4, CO2, CO and H2O." l.106-107

**and**

"Also on board the ATR42 were two Picarro™ models. One was the previously mentioned G2401 and the other was the G5310 that measures CO with higher precision as well as N2O. Other instruments installed on the aircraft also gathered additional meteorological data." l. 121-124